# Accelerated western European heatwave trends linked to more-persistent double jets over Eurasia

Efi Rousi [1✉], Kai Kornhuber [1,2,3], Goratz Beobide-Arsuaga [4,5], Fei Luo [6,7] & Dim Coumou [1,6,7]

Persistent heat extremes can have severe impacts on ecosystems and societies, including excess mortality, wildfires, and harvest failures. Here we identify Europe as a heatwave hotspot, exhibiting upward trends that are three-to-four times faster compared to the rest of the northern midlatitudes over the past 42 years. This accelerated trend is linked to atmospheric dynamical changes via an increase in the frequency and persistence of double jet stream states over Eurasia. We find that double jet occurrences are particularly important for western European heatwaves, explaining up to 35% of temperature variability. The upward trend in the persistence of double jet events explains almost all of the accelerated heatwave trend in western Europe, and about 30% of it over the extended European region. Those findings provide evidence that in addition to thermodynamical drivers, atmospheric dynamical changes have contributed to the increased rate of European heatwaves, with implications for risk management and potential adaptation strategies.

[1] Potsdam Institute of Climate Impact Research (PIK), Member of the Leibniz Association, Potsdam, Germany. [2] Earth Institute, Columbia University, New York, NY, USA. [3] Lamont-Doherty Earth Observatory, Columbia University, New York, NY, USA. [4] International Max Planck Research School on Earth System Modelling, Hamburg, Germany. [5] Institute of Oceanography, Center for Earth System Sustainability, Universität Hamburg, Hamburg, Germany. [6] Institute for Environmental Studies, Vrije Universiteit Amsterdam, Amsterdam, Netherlands. [7] Royal Netherlands Meteorological Institute (KNMI), De Bilt, Netherlands. ✉email: e.rousi@gmail.com

Heat extremes have increased on a global scale over recent decades and are expected to further increase under future global warming[1–3]. Europe has seen a particularly strong increase in heat extremes since the deadly summer 2003 heatwave[4,5], which is estimated to have caused ~70,000 excess deaths[6]. This tendency is illustrated by the recent cluster of consecutive exceptionally hot and dry summers of 2018[7], 2019[8] and 2020. European heatwaves are projected to increase disproportionately compared to the global mean temperature in the future[9] but the underlying reasons are not well understood.

Drivers of European summer hot temperatures and heatwave variability include large-scale atmospheric circulation and jet stream states[10–13], soil moisture deficit and related land-atmosphere feedbacks[14–16], oceanic circulation and sea-surface temperatures[13,17]. Anthropogenic global warming, mainly due to increasing GHGs, increases the intensity and frequency of heatwaves by direct warming[18,19] but can also affect these drivers of natural variability[20].

Observational[21] and model-based[22] studies have shown that summer heat extremes over the northern midlatitudes are primarily associated with blocking anticyclones. In turn, those blocking high pressure systems are often linked to a double jet stream structure over Eurasia that favors their formation in the region of weak winds between the two maxima in the zonal wind[11,23,24]. Alternatively, Rossby wave-breaking and consequent blocking may also cause the split of the jet stream and the occurrence of double jets. Either way, the existence of a double jet in the troposphere is characterized by a very confined subtropical jet that can affect Rossby waves in the midlatitudes favoring the stagnation of ridges and troughs[11,25]. Accelerated high-latitude land warming during boreal summer, which has been attributed to anthropogenic climate change, could provide favorable conditions for the occurrence or the persistence of double jet states, via strengthening of the polar jet front[26]. Rossby wave theory also suggests that double jet flow regimes can become slightly more common as the zonal flow weakens under pronounced Arctic Amplification[27].

Still, there is little evidence for changes in the frequency and intensity of summer European blocking under historical or future global warming[28], which constitutes a discrepancy with the increasing trend in European heatwaves[29]. However, modeling studies have reported an anomalous high-pressure response located off the UK coast in future warming scenarios in summer, favoring hot and dry weather over western Europe[30,31].

Here, we study how European temperature extremes are linked to large-scale atmospheric circulation and in particular jet stream states and analyze how potential changes therein might have contributed to upward heatwave trends. We argue that the accelerated trend in western European heatwaves is linked to an increase in the persistence of double jets in the upper troposphere.

## Results

**Amplified heatwave trends over Europe.** Trends of heatwave frequency and cumulative intensity have increased over many regions of the midlatitudes, with Europe among those with the most pronounced trends (Fig. 1a, b; see Fig. S1a, b for plots with the grid point-level statistical significance of the trends). This holds particularly true for persistent heatwaves, which are defined here as at least 6 consecutive days of temperature threshold exceedance (see Methods and see Supplementary Material Figs. S2, S3 based on heatwaves for a 3-day exceedance). Cumulative intensity[3] refers to the sum of the excess heat (above the 90th percentile of maximum temperature) for all heatwave events in the high-summer season (here defined as

July-August) for each grid point (see Methods). Heatwave frequency and cumulative intensity of heatwaves show a very similar trend pattern, as cumulative intensity is proportional to the number of heatwave days accounted for. Both heatwave metrics are increasing over almost all regions in the northern hemisphere (NH) midlatitudes (Fig. 1a, b), with the main exceptions being central North America (known in the existing literature as the US "warming hole"[32,33]), central Siberia, and northern India (probably attributable to local cooling due to intensive irrigation and aerosols over the Indo-Gangetic Plain[34]). Hotspots with a particularly pronounced increase in heatwaves are seen over Europe, the Middle East, parts of China, and western North America, in agreement with Perkins-Kirkpatrick and Lewis[3].

Both frequency and cumulative intensity of heatwaves show a much faster increasing rate in Europe compared to the rest of the midlatitudes for the large majority of land grid points (Fig. 1c, d). In particular, heatwave days show a mean increase over Europe of +0.61 days/decade, compared to +0.21 days/decade for the rest of the midlatitudes (Fig. 1c), constituting a ~3 times faster rate for Europe. This accelerated increase for Europe is even more pronounced when looking at heatwave cumulative intensity, which shows a ~4 times larger trend compared to the rest of the midlatitudes (Fig. 1d). The distributions of decadal trends of both heatwave metrics of the two regions differ significantly (0.99 level of significance) according to a Kolmogorov-Smirnov statistical test.

**Jet stream states and associated Eurasian temperature anomalies.** Changes in the large-scale atmospheric circulation and the jet stream can affect the spatiotemporal variability of heat extremes. Here, we employ a neural network-based clustering algorithm, Self-Organizing Maps (SOMs)[35], to identify dominant jet stream states during boreal high summer months (July-August, see Methods). Three clusters of jet states are objectively identified for the historical period based on the vertical structure of the zonal-mean zonal wind (u, see Methods): a pronounced single jet stream state (Fig. 2a), a clear double jet structure (Fig. 2b), and a mixed jet pattern (Fig. 2c). As a test of robustness, we applied this analysis on the longer NH warm season (May-September) confirming these three dominant states and also showing that the double jets occur almost exclusively in July and August (Figs. S4, S5).

In July-August, the cluster of single jet stream states is the most frequent (37% of all days) and shows a slightly decreasing trend (not statistically significant) in annual frequency and persistence (Fig. 2d). The wind maximum, at a pressure level between 250-150hPa, corresponds to an enhanced zonal jet stream blowing over the whole latitudinal band of central/northern Eurasia, directly north of the climatological jet stream (~45 to 60°N, Fig. 2g). The composite of near-surface temperature for the days classified in this cluster shows higher temperatures over southern Europe and central Asia (Fig. 2j), essentially over the whole latitudinal band that is located southwards of the single pronounced jet stream. Further, single jets are linked to mildly increased heatwave cumulative intensity over parts of the Mediterranean (Fig. 2m).

Double jets are almost as frequent as single jets (36% over the full period; Fig. 2b) and show significant upward trends both in frequency (~3 days/decade) and persistence (~2 days/decade; Fig. 2e). Double jets are characterized by two maxima of the Eurasian zonal-mean zonal wind in the vertical structure of the troposphere with a minimum in between (Fig. 2b). Composites show two zonally oriented bands of strong positive wind anomalies at the 250hPa level (Fig. 2h) that occupy the whole North Atlantic/Eurasian domain. The strongest wind maximum

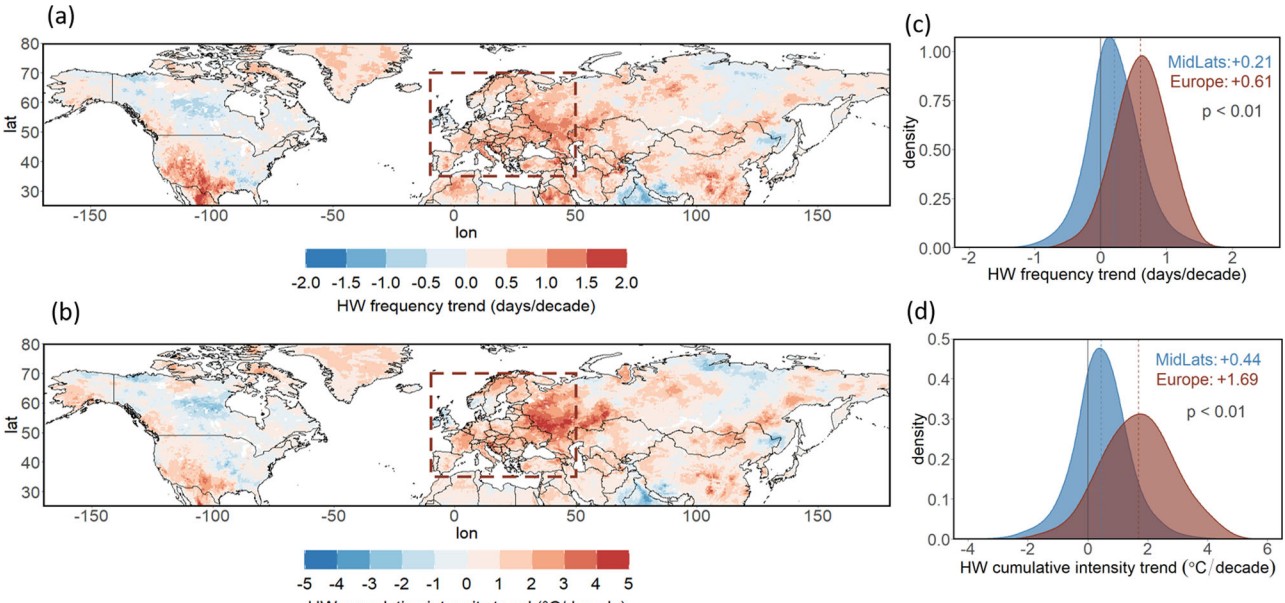

**Fig. 1 Increasing heatwave trends over the midlatitudes and Europe. a** Decadal trends in heatwave frequency (days/decade) and **b** heatwave cumulative intensity (°C/decade) for July-August 1979–2020. **c** Probability density distributions of decadal trends of heatwave frequency of all land grid points for Europe (in dark red, as the region included in the dashed box of (**a**, **b**): 35–70°N and 10oW-50°E) and the midlatitudes (20–70°N) excluding Europe (in blue) and **d** probability density distributions of decadal trends of heatwave cumulative intensity. The mean trend for each distribution is shown with dashed vertical lines and provided on the top right of the panels. The continuous vertical lines correspond to 0 (i.e. no trend). The two distributions were compared for each case with a Kolmogorov-Smirnov test (*p* values shown on the center-right).

represents an enhanced and confined subtropical jet stream, centered at around 40°N, and an Arctic front jet at ~70°N, seen as a prominent band of positive anomalies of the zonal wind over the higher latitudes. The mid-latitudinal belt located between the two jets (~45–65°N) is characterized by negative wind speed anomalies and particularly weak winds with anomalously warm surface temperatures (Fig. 2k). Heatwave cumulative intensity particularly increases, as compared to the climatological mean, over this mid-latitudinal belt, affecting large parts of Eurasia. The most pronounced peak is seen over western Europe (Fig. 2n), but some other pronounced anomalies are seen further east, e.g. Russia and western and eastern Siberia.

Thus, double jets are linked to heat extremes over western Europe (and also the Baltic region), while single jets are linked to heat extremes over part of the Mediterranean, but to a lesser degree. This is consistent with physical considerations, as the single jet is actually a northward shifted jet stream that allows for heat extremes to develop in southern regions (bellow ~45°N). On the other hand, with the double jet configuration, the wind minimum over the latitudinal zone of 45–65°N favors persistent high-pressure weather systems there, including blocking anticyclones, impeding the propagation of synoptic disturbances and therefore increasing the probability of long-lasting heatwaves. The mixed jet state (Fig. 2c) features a localized double jet structure over North Atlantic, which is not well-defined over the whole Eurasian sector (Fig. 2i), and it is less relevant for European heat extremes (Fig. 2l, o). From this point on we will focus our analysis on the double jet states.

Next, we analyze the role of meridional wind anomalies (v250) in favoring heat extremes in specific locations during the most-persistent double jet events (those exceeding the 90th percentile of persistence). The meridional wind composite at the 250hPa level for the most-persistent double jet events shows an amplified circumglobal wave pattern (Fig. 3a), which is similar to the one highlighted as important for heat extremes in western Europe in previous research[7,36,37]. Then, by further

clustering the meridional wind field of those persistent double jet events, we show that this composite originates from two preferred wave patterns (Fig. 3b, c). Hence, double jets are associated with amplified waves that come in two preferred positions over Eurasia that are phase-shifted by half a wavelength. SOM1 represents northerly winds over Scandinavia/western Russia and southerlies over the Ural Mountains (Fig. 3b) creating heat extremes over Russia (Fig. 3f). This pattern shows a statistically significant upward trend (Fig. 3d), based on a similarity index of its composite with the daily v250 wind fields (see Methods). In SOM2 this pattern is shifted to the west (Fig. 3c) favoring heat extremes in western Europe (Fig. 3g). The similarity index of SOM2 shows a small and non-significant downward trend (linear), which however turns to an upward trend in the latest decade when applying a non-linear fitting (2010–2020; Fig. 3e). These European wave-induced anomalies are part of a larger circumglobal wave pattern creating warm anomalies in western (SOM2) and eastern Siberia (SOM1). The anomaly composites of mean surface temperature for those two clusters (Fig. 3f, g) show 4 hotspot regions (Russia and eastern Siberia for SOM1, and western Europe and western Siberia for SOM2), in agreement with the hotspot regions showing enhanced heatwave risk during double jets (Fig. 2n).

The most persistent double jet configuration was seen in the summer of 2003 (see Table S1 in the Supplementary Material for a list of the 20 most persistent double jet events in the period studied), which also saw one of the strongest European heatwaves of the observational era[9], resulting in ~70,000 heat-related deaths[6]. Figure 4 showcases 4 extreme summers, in terms of both Eurasian double jet persistence and heatwave occurrence and intensity in central/western Europe, i.e. 1994, 2003, 2006[9], and 2018. The first column (panels a,d,g,j) shows the mean anomalies of the zonal wind at 250hPa for the period of the most-persistent double jet event for each of those summers. In all cases, the zonal wind is significantly weaker

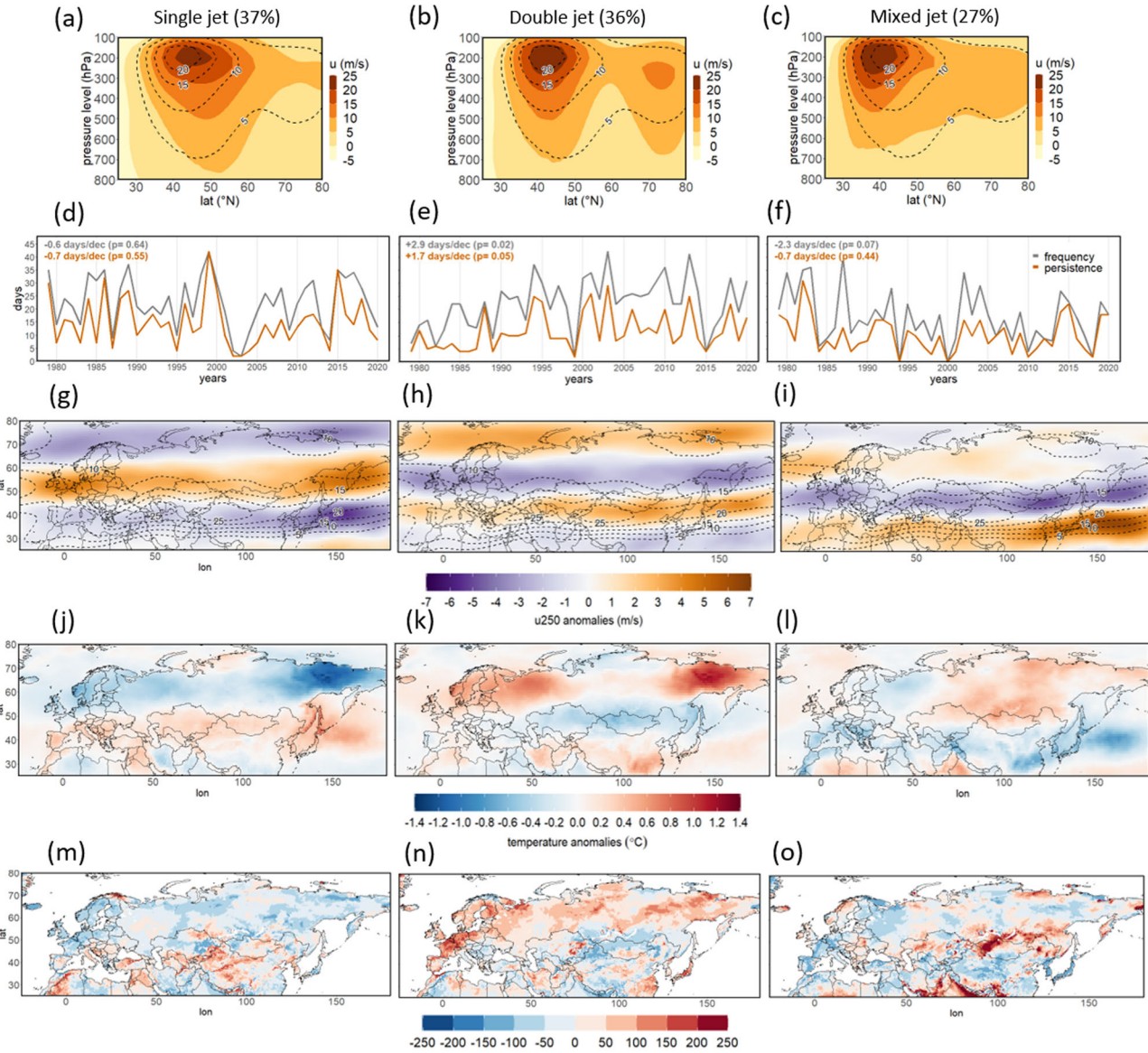

**Fig. 2 Jet stream states and surface temperature. a–c** Clusters of the vertical profile of the zonal (averaged over the Eurasian domain) mean zonal wind (u, shading) with frequency of occurrence provided in parenthesis. The climatological mean of the zonal mean zonal wind for the whole period is plotted with dashed contours (plotted from 5 to 20 m/s every 5 m/s). **d–f** Frequency (gray line) and maximum persistence (orange line) of each cluster per year. The decadal linear trend and respective p-values are given for both time series on the top left of the panels. **g–i** Anomaly composites of (linearly detrended) zonal wind at the 250hPa pressure level (u250) for each cluster (shading). The climatological mean of the zonal wind at 250hPa is plotted with dashed contours (plotted from 5 to 25 m/s every 5 m/s). **j–l** Anomaly composites of (linearly detrended) mean surface temperature for each cluster. Anomalies in both cases are calculated with respect to daily climatology (to remove the seasonal cycle). **m–o** Composites of heatwave cumulative intensity (calculated after having removed the Tmax mean midlatitude-land trend from each grid point) shown as relative anomaly (%) compared to the climatology. All figures refer to the months of July-August of the period 1979–2020.

over continental Eurasia, while two belts of stronger winds prevail over southern and northern latitudes, constituting the distinctive double jet configuration. Focusing on July-August of 2003, that show the most-persistent double jet event of the whole study period (Table S1), we can see that double jets were first established in the beginning of July (as seen in the Eurasian zonal mean zonal wind presented in 5-day running means in the Hovmöller diagram of Fig. 4e) and a heatwave developed afterwards on the 12th of July over central France. The spatial extent of the cumulative intensity of the 2003 heatwave shows a peak over this area but also covers parts of Spain, Italy, the UK, most of Germany, and the Low Countries (Fig. 4f). Note that the spatial pattern of the 2003 heatwave bears great similarity to

the composite pattern of cumulative heat anomaly during double jet events, including a second warm anomaly in Baltic regions (as seen in Fig. 2n). The 2006 and 2018 heatwaves (Fig. 4i, l) also have a similar spatial pattern, while in 1994 (Fig. 4c) the heat is shifted towards a more eastward location. As seen in the climatology of the zonal wind at the 250hPa level (dashed contours in Fig. 4a, d, g, j) the western/central European region is normally coinciding with the exit region of the North Atlantic/polar jet stream. When we have double jets, this region is characterized by negative anomalies of the wind, while positive anomalies (corresponding to the 2 jets) can be seen to the north and to the south of the region.

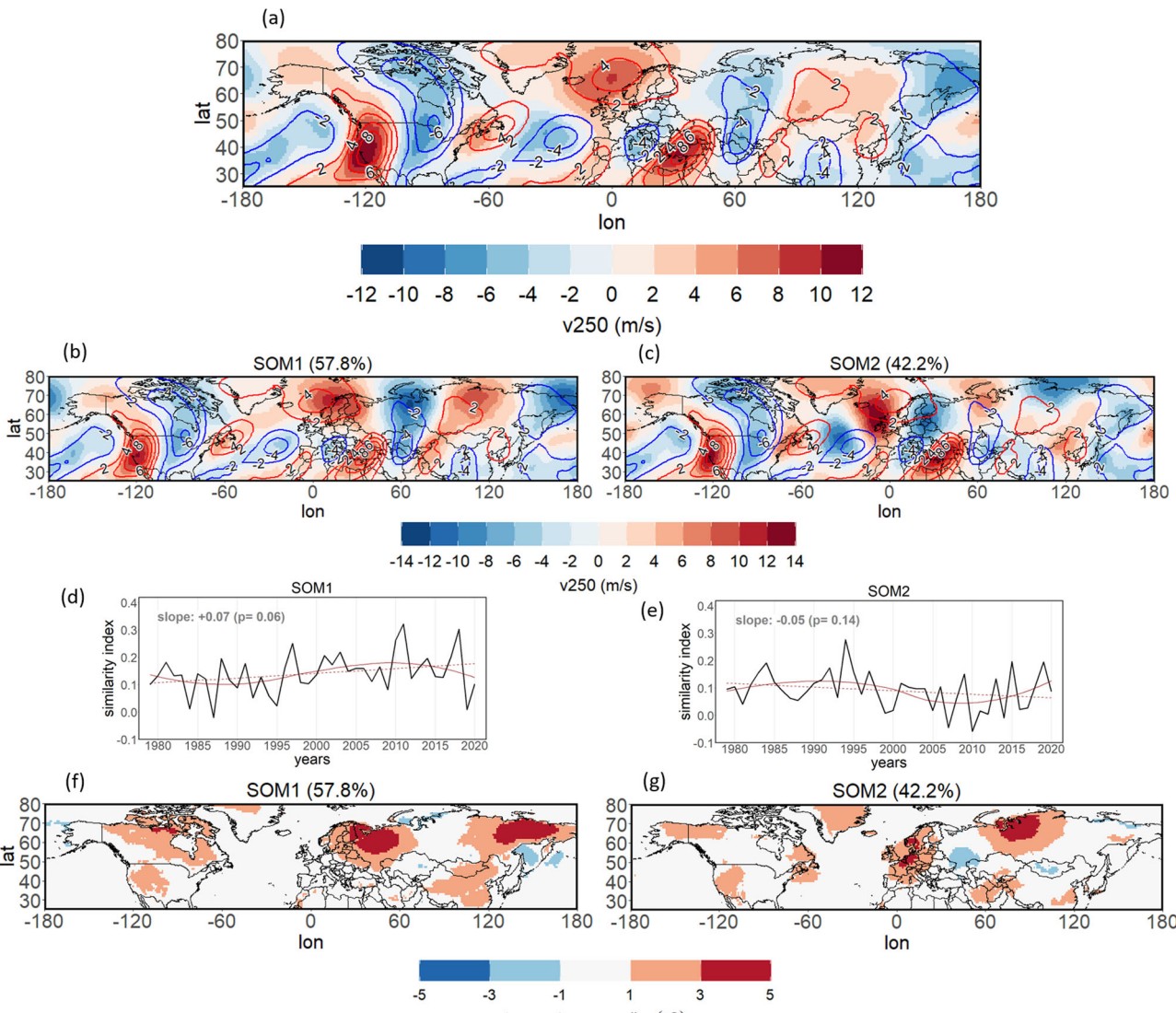

**Fig. 3 Persistent double jets states. a** Composite of meridional winds at 250hPa (v250) for double jet events exceeding the 90th percentile of double jet persistence (i.e, events lasting more than 11 consecutive days, see Table S1 for the 20 most persistent events and their duration; shading). Contour lines show the v250 July-August climatology for the whole period 1979–2020 (plotted from −5 to 8 m/s every 2 m/s). **b, c** SOM cluster composites of v250 for persistent double jet events (frequency % of each SOM is shown in parenthesis). **d, e** Time series of daily similarity index for the two SOMs of v250 winds. The red dashed line shows a linear regression fit, with its slope and p-value plotted on the top left. The continuous red line shows a smoothed LOESS curve fit (span of 0.75). **f, g** Anomaly composites of (linearly detrended) mean surface temperature for each of the SOM clusters.

**Contribution of double jets to European heatwave variability and trend.** Double jets explain up to 35% of heatwave variability over parts of western Europe and their upward trend has contributed significantly to the observed amplified European heatwave trend. Using linear regression analysis (see Methods), we show that double jet persistence explains a large part of the variability in European heatwave cumulative intensity (Fig. 5). To account for potential biases due to the seasonal cycle and long-term trends, we first detrended both the regressor (double jet persistence) and the response variable (heatwave cumulative intensity) before applying the regression analysis. As seen in Fig. 5a, double jet persistence explains up to 35% of the variability of heatwave cumulative intensity over parts of western Europe, a region stretching from Spain to the Baltic countries. Similar, albeit lower, coefficients are obtained when using double jet frequency (instead of persistence, see Fig. S6), or when regressing on heatwave frequency (instead of cumulative intensity, see Figs. S7, S8). To further investigate the relation between double jets and

cumulative heat in Europe we plot the double jet persistence against the spatially averaged cumulative heat over Europe (Fig. 5b) and western Europe in particular (Fig. 5c). In both cases we find positive linear relationships that are much more pronounced and become statistically significant when focusing on western Europe, as seen also in Fig. 5a. The mean explained variance is 5% for the whole European domain, while it reaches 24% for western Europe (Fig. 5c). Those findings supplement the results from the composites of heat extremes (Fig. 2n) confirming that persistent double jets are particularly relevant for western European heatwaves.

Increasingly persistent double jets can explain up to ~1/3 of the accelerated European heatwave trend and almost all of the accelerated trend over western Europe (Fig. 6). The heatwave cumulative intensity trend at each grid point was estimated using the linear regression model based on the double jet persistence alone (see Methods). We assume that the direct thermodynamic contribution to heatwave trends is of similar magnitude across

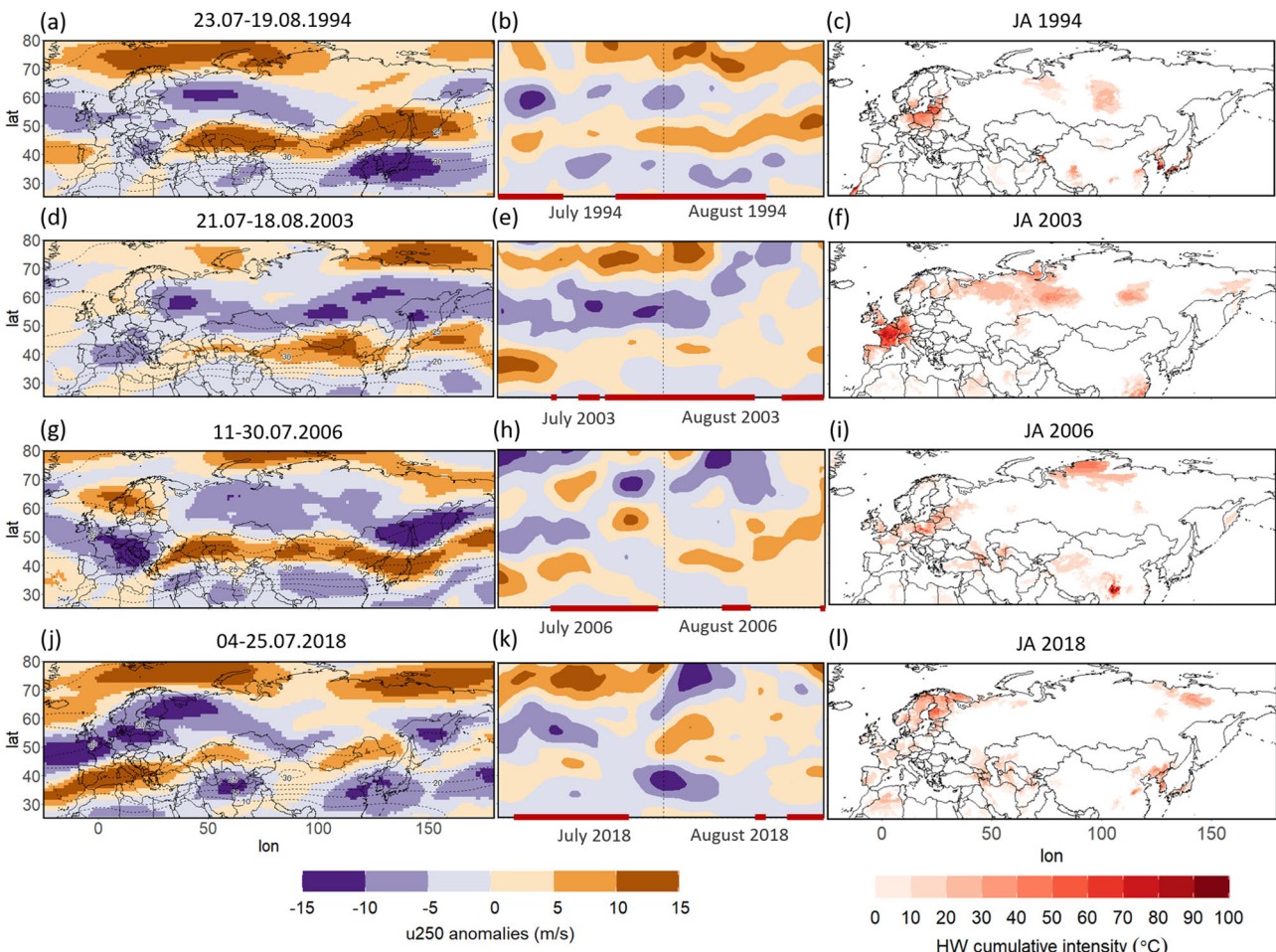

**Fig. 4 Summers 1994, 2003, 2006, 2018: double jets and heatwave intensity. a** Anomalies of the 250hPa zonal wind (u250; shading, anomalies from climatology; dashed contour lines show the total wind speed climatology plotted every 5 m/s) for the longest double jet event in 1994: 23.07-19.08.1994. **b** Hovmöller diagram of the Eurasian (region seen in panel a) zonal mean zonal wind anomalies for July-August 1994 (shading; 5day running means centered on each day from 01.07-31.08.1994). The vertical dashed line refers to the first day of August and the red horizontal lines on the time axis show the days identified by the SOMs as double jets (dates for 1994: 1-12.07, 23.07-19.08). **c** Spatial distribution of heatwave cumulative intensity for July-August 1994. **d** As in **a** but for 21.07-18.08.2003. **e** As in **b** but for July-August 2003 (dates of all double jets for 2003: 11.07, 16-19.07, 21.07-18.08, 24-31.08). **f** As in **c** but for July-August 2003. **g** As in (a) but for 11-30.07.2006. **h** As in **b** but for July-August 2006 (dates of all double jets for 2006: 11-30.07, 12-16.08, 31.08). **i** As in **c** but for July-August 2006. **j** As in (a) but for 04-25.07.2018. **k** As in (b) but for July-August 2018 (dates of all double jets for 2018: 04-25.07, 19-20.08, 25-31.08). **l** As in **c** but for July-August 2018.

the midlatitudes. This is a first-order approximation giving an estimate of the thermodynamic contribution by the mean midlatitude trend of 0.62 °C/dec. For Europe, we then define a "residual trend" as the total trend minus this thermodynamic contribution, which averaged over the full European region is ~1.05 °C/dec, and ~0.54 °C/dec for western Europe (Fig. 6a). The residual trend is thus the enhanced trend over Europe, as compared to all midlatitudes, and we assume this residual trend to be attributable to more complex processes like feedbacks or dynamical changes. Our linear model captures fairly well the decadal trend patterns over large parts of Europe, but with smaller magnitudes (Fig. 6b, note difference in scale compared to 6a). This is to be expected as other factors, such as soil moisture-temperature feedbacks[15], certainly play a role in shaping or amplifying heatwave trends. Still, the increasing persistence of double jet states captures large parts of the residual trend, especially over western Europe and European Russia. Following this approach we find that, averaged over Europe, ~30% of the total observed trend in HW cumulative intensity can be attributed to the increase in double jet persistence, under the assumptions provided above. Even more strikingly, for western Europe the

contribution is much higher, i.e. ~100%, thus explaining all of the residual trend. This supports the hypothesis that for western Europe the dynamical changes towards more-persistent double jets are key to understanding the accelerated heatwave trend.

## Discussion

A possible driver of more-persistent double jet events is the increased thermal contrast across the Arctic coastline due to the enhanced high-latitude land warming compared to essentially no warming over the cooler Arctic ocean. The Arctic ocean has seen little or no warming in summer, as all additional energy from greenhouse gas forcing is used to melt sea ice and ocean surfaces generally show a slower warming trend. In contrast, the land area surrounding the Arctic ocean (Siberia, Alaska, Canada, etc) has seen very rapid warming in summer, likely also linked to stark reductions in early summer snow cover[38]. This implies that while the overall equator-to-pole temperature gradient is reducing due to Arctic Amplification[39], the thermal gradient increases north of the Arctic circle, which strengthens the Arctic front jet (at ~70-80°N)[25]. Mann et al.[26] showed that the historical zonal-

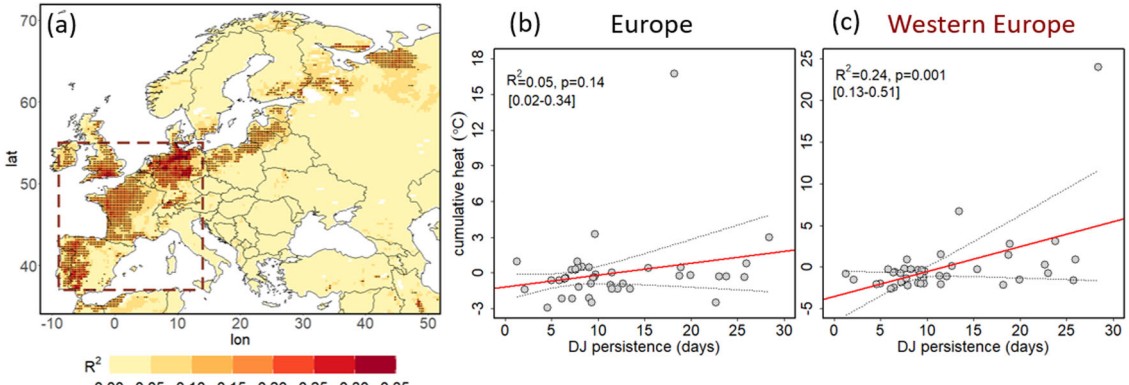

**Fig. 5 Explained variance of heatwave cumulative intensity by double jet persistence. a** Explained variance ($R^2$) per grid point of heatwave cumulative intensity based on linear regression on double jet persistence. Statistically significant coefficients ($p < 0.05$) are marked with black dots. **b** Scatter plots of heatwave cumulative intensity anomalies aggregated over all land grid points of the extended European domain (as seen in panel a and in the red dashed box of Fig. 1a, b) and double jet persistence. A linear fit (in red) and its confidence interval (dashed lines for the 5th and 95th percentiles obtained from 1000 bootstraps), $R^2$ (with 5th and 95th interval percentiles obtained from 1000 bootstraps in brackets), and p-values are shown on the top left of each plot. **c** As for **b** but with heatwave cumulative intensity aggregated only over land grid points with statistically significant coefficients in western Europe (dotted points in panel a within the region included in the dashed red box: 37–55°N and 9°W–14°E). The linear trend of the time series was removed before the regression was applied in all cases.

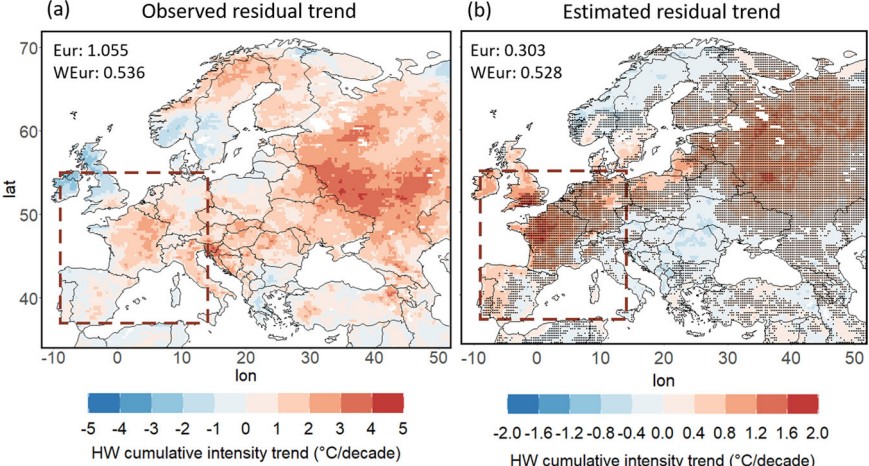

**Fig. 6 Observed versus estimated heatwave cumulative intensity trends based on the increase of double jet persistence. a** Observed residual trend (observed trend—mean midlatitude-land trend) in heatwave cumulative intensity (°C/decade) over Europe. **b** Estimated residual trend in heatwave cumulative intensity (°C/decade) over Europe. Grid points for which the estimated residual trend is of the same sign with the observed residual trend are marked with dots. Mean residual trend (°C/decade) for Europe and western Europe (region included in the dashed red box) is given on the top left of each panel.

mean surface warming profile, characterized by accelerated warming over high-latitude land areas in summer, is likely ('likely' following the IPCC lexicon) attributable to anthropogenic climate change. We argue that this zonal-mean warming pattern favors more-persistent double jet flow regimes. Thus, while double jets might be triggered first of all by chaotic dynamics in the midlatitudes or via tropical Rossby wave forcing[40,41], their persistence increases in a climate with much warmer land areas in the Arctic. Our results also link to previous work on quasi-resonant amplification[42], which leads to high-amplitude circumglobal waves, and is often associated with double jet flow regimes. The double jet configuration can provide the necessary latitudinal waveguide via a confined subtropical jet[43] that traps and amplifies Rossby waves. Such states have been identified for extreme summer months[44]. As we show in our analysis, double jets are associated with a strong and confined subtropical jet (favoring waveguidability) and amplified circumglobal wave patterns in the meridional wind. These patterns bear great

similarity with the preferred position of the circumglobal wave pattern that has been found to be important for heat extremes in western Europe in previous research[7,36,37]. Likewise, such confinement is also possible for the polar jet in boreal summer[45]. Alternatively, the waveguidability diagnosed in the zonal-mean zonal winds may also be a consequence of - and not a precondition for - large wave amplitudes[46].

Our work adds to the body of evidence that double jets and blocking are strongly related. The exact causal relationships are more difficult to interpret. On the one hand, one can argue that a double jet configuration leads to the formation of a blocking anticyclone in the region of weak winds between the two zonal-wind maxima. Tachibana et al.[23] found that an anomalously positive Northern Hemisphere Annular Mode (NAM), characterized by a double jet, accounts well for the hemispheric-scale weather associated with anomalous blocking in the regions between the two jets. In that case, the double jet stream tends to cause atmospheric blocking, in agreement with Maeda et al.[47].

Additionally, a double jet could favor the maintenance of a high-latitude blocking anticyclone by advecting low potential vorticity air into the system[48]. However, a reverse causal chain can neither be excluded; a blocking anticyclone over e.g. western Europe could make the jet stream split in two flanks and thereby create a double jet state[49]. Double jets may thus both be a cause for, or a consequence of blocking. Understanding causal chains requires further research, alongside investigating local Rossby wave activity, such as transient Rossby wave packets and temperature extremes[50]. Our preferred interpretation is that double jets and blockings are two aspects of the same dynamical flow pattern that can be triggered by different processes, including internal atmospheric dynamics (e.g. resonance effects), ocean-atmosphere interactions[51], or tropical Rossby wave forcing[41]. Summer changes in the Arctic, as described above, are now making such dynamical states more stable and thereby favor more-persistent double jet events. This mechanism needs to be tested with the use of climate model experiments with different Arctic conditions scenarios and with existing CMIP6 projections, which is topic of our future work. Furthermore, it is important that other mechanisms that have been discussed in recent studies, such as the role of suppressed tropical Pacific convection in enhancing jet stream waviness[41], or the role of the differential aerosol forcing in the weakening of the Eurasian jet[52], are reconciled with the role of the Arctic and brought together in future studies.

Double jets were found to be particularly important for heatwave variability and trends in western Europe (Figs. 4, 5). In this region the jet stream is playing an active role in modulating surface conditions as it coincides with the exit region of the North Atlantic storm track[53]. In other European regions, such as the Mediterranean and eastern Europe, local land-atmosphere feedbacks might be more important than jet stream dynamics due to their strong coupling and the impact of soil preconditioning[54]. Land atmosphere-feedbacks from desiccated soils are important drivers of heatwaves[14]. In that context, Stegehuis et al.[55], based on regional climate model simulations for the historical period, find that early summer soil moisture explains more than half of the warming trend over France and southwestern Germany, while large-scale drivers are dominant over the rest of Europe. Our results are generally consistent with this, suggesting however a more significant contribution of large-scale dynamics to heatwaves over western France, which could be explained by the fact that we focus on long-lasting heatwaves, while they look at summer mean temperatures[55]. Finally, several studies[30,31] have discussed the potential role of an AMOC slowdown and associated North Atlantic warming hole[56] for western European summer climate, mediated by an ocean-induced atmospheric response. Although the exact processes, including atmosphere-ocean interactions, are not well understood, nor adequately captured in climate models[57], the atmospheric response to an AMOC slowdown in summer is characterized by more-frequent high-pressure over western Europe[30,31]. Future heatwave risks for Europe are thus likely governed by the atmospheric response to different boundary forcings, including high-latitude warming, an Atlantic cold anomaly, and generally lower soil-moisture conditions.

This is the first time, to the best of our knowledge, that the accelerated European increase in summer persistent heat extremes is quantified and linked to specific dynamical changes in the jet stream. Future research should investigate how well climate models capture the links between heat extremes and double jets and whether any of those components change under different forcing scenarios. Such analysis would also allow to assess whether the observed increase in double jets is part of internal natural variability of the climate system or a response to anthropogenic climate change[58]. Our findings and further analysis could help improve climate models that are currently underestimating the observed warming trend over western Europe[55,59]. If models do not accurately represent the variability of the jet stream this could result in a significant underestimation of future heatwave trends over western Europe. Certain jet stream configurations are also linked to concurrent extremes in different midlatitude regions that can pose simultaneous risks in multiple breadbaskets, endangering global food security and social stability[37]. Therefore, a better understanding of their driving forces and implications is crucial for more robust risk assessments under unmitigated climate change.

## Methods

**Data**. To study jet stream states over Eurasia we used ERA5 reanalysis data[60] for the zonal-mean zonal wind (u) over the area 25°−80°N, focusing on the Eurasian sector (25°W−180°E), for the pressure levels 800hPa-100hPa, and for the months July and August. The size of the domain and the pressure levels were tested and the results were found to be insensitive to changes (e.g. 0–90°N, or pressure levels 1000hPa-100hPa). The data are in a 0.28° × 0.28° spatial resolution and daily values were calculated from 6-hourly data (in particular from the timesteps 00:00, 06:00, 12:00 and 18:00). Other variables used and presented in composite maps, such as mean and maximum surface temperature (used to calculate the heatwave metrics, see "Heatwave definition, metrics, and trends") and zonal (u) and meridional (v) wind at the 250hPa pressure level were also retrieved from the ERA5 datasets.

**Heatwave definition, metrics, and trends**. Although there is no universal definition for heatwaves, there are some criteria and thresholds that have been used and tested extensively in the literature. Main characteristics of a heatwave are the physical one which describes its intensity, referring to the temperature ranges reached, the temporal that refers to its duration, and the spatial, which describes the spatial extent of a heatwave. Here we define a heatwave day based on the following criteria:

- Temperature threshold: the daily maximum temperature has to exceed the 90th percentile of the maximum temperature distribution of the period studied based on a centered 15-day window (Tmax >90th percentile), following Fischer and Schär[61].
- Temporal extension: we define heatwaves as at least 3[62] or 6[61] consecutive days of temperature threshold exceedance. In the main manuscript we present the results referring to long heatwaves (≥6 consecutive days), while results for shorter heatwaves (≥3 consecutive days) are presented in the SI.
- Spatial extent: we define an event as a heatwave if it exceeds an area of 40.000 km² within a 4° × 4° sliding window (similar to Stefanon et al.[63]). Different sliding windows were tested and did not have a significant effect on the heatwave detection.

Apart from the heatwave frequency we are also interested in the heatwave intensity. Here we look at the heatwave cumulative intensity, as defined by with Perkins-Kirkpatrick and Lewis[3], which refers to the integration of heat exceedance over the threshold for each heatwave event. When referring to a whole region, we additionally aggregate the heat exceedance for the whole spatial extent:

$$heatwave\ cumulative\ intensity = \sum_1^{gp} \sum_1^{d} (Tmax - Tmax_{90th}) \qquad (1)$$

where gp is the number of land grid points for each region, d is the number of consecutive heatwave days, Tmax the maximum daily temperature and $Tmax_{90th}$ the 90th percentile of the maximum temperature distribution of the whole time period 1979–2020.

The heatwave cumulative intensity is a useful metric as it integrates in a single number all the characteristics of a heatwave: intensity, duration and spatial extent. This way it enables easier comparisons between different regions or years. Additionally, considering the excess heat experienced once the heatwave threshold is exceeded makes this metric more impact-relevant[3].

In all cases where heatwave cumulative intensity is aggregated for a certain region, only the land grid points are considered and the data are weighted by the cosine of the latitude to account for the different size of the grid cells between different latitudinal zones.

All trends reported in this study, such as the trends of the heatwave metrics and those of the double jet frequency and occurrence, are simple linear trends calculated as the slope of the least squares line of each variable against time. Additionally, a LOESS regression smoothing has been applied in the case of the trends of the SOM similarity indices in Fig. 3d, e. For the comparison of distributions, the Kolmogorov-Smirnov test was used.

**Identification of jet stream states**. In order to detect double jet configurations in the vertical zonal-mean zonal wind field, we used a neural network-based, unsupervised clustering algorithm, Self-Organizing Maps (SOM[35,64]). SOMs have been used a lot in recent decades in atmospheric sciences[31,65] and they provide a flexible alternative to other clustering algorithms, such as k-means and hierarchical

clustering. The SOM algorithm starts with randomly initialized weight vectors (c), followed by a sampling step, during which a random input vector ($x$) is compared to all weight vectors until its Best Matching Unit (BMU) is detected according to a distance metric (here we use sum of squares). Next, the BMU and its neighboring units are updated to become more similar to the newly added input vector, according to the following function:

$$\mathbf{c}_k(t+1) = \mathbf{c}_k(t) + \alpha(t) \times h_{ck}(t) \times [x(t) - \mathbf{c}_k(t)] \quad (2)$$

where $c_k$ the BMU weight vector, $\alpha$ the learning rate parameter that decreases with each iteration t, and $h_{ck}$ a neighborhood function determining how many neighbor nodes surrounding the BMU will be affected. Here, the bubble type neighborhood function was used:

$$h_{ck}(t) = F(\sigma(t) - d_{ck}) \quad (3)$$

where $\sigma$ is the neighborhood radius that decreases linearly with iteration (t) until it reaches 0, when no neighbor nodes are updated anymore, $d_{ck}$ the Euclidean distance between the BMU (c) and another one of the SOM nodes (k), and F(x) is a step function that takes the value 1 as long as the neighborhood radius remains larger than the Euclidean distance, and the value 0 when the radius becomes equal to it.

This procedure is repeated for each one of the input vectors (the algorithm goes back to the sampling step) until the final SOM array does not change anymore. The SOM finalization is achieved when all input vectors have been assigned to their BMU. The neighborhood function included in the SOM algorithm is the element that makes this method different from other clustering techniques, such as k-means, as it allows for a topological ordering of the SOM clusters in the final SOM array, which represents the structure of the input data.

One of the choices that has to be made a priori with SOMs is how many clusters will be employed and this heavily depends on the scope of the study and the application and the degree of generalization that one wants to achieve. For our analysis, we tried different SOM sizes (from 2 to 6, not shown) and found that 3 SOMs represent the different jet stream states that we are interested in to a good degree. More than 3 SOMs produced very similar clusters, while 2 SOMs resulted in too general patterns. Additionally, we tested k-means and hierarchical clustering on the same data and with the same number of clusters (2–6) and the results were fairly similar (not shown).

Furthermore, due to the inherent stochasticity of the SOM algorithm, slightly different results may be obtained from different random initializations and therefore multiple random runs are recommended[66]. Here we use 10 random runs and choose the SOM array that better represents the input data according to the quantization error that measures the goodness of the final SOM in terms of similarity of the SOM clusters to the contained data vectors[67]:

$$quantization\ error = \frac{1}{N}\sum ||\vec{x}_i - \mathbf{c}_{\vec{x}_i}|| \quad (4)$$

where N is the number of data vectors $x$ and c its best matching unit, i.e. the weight vector of the SOM cluster in which it is classified.

Apart from testing different numbers of SOM clusters and running 10 random initializations, we performed sensitivity analysis regarding the months used as input for the SOMs, and the spatial domain. Initially, June-July-August (JJA) data were tested, but the results showed that June has a fairly different behavior compared to July and August in terms of the jet stream, being more of a transitional month between spring and summer. This might be related to the fact that the most pronounced thermal gradient over the Arctic circle is seen in high-summer months, due to the largest differences in land versus sea warming[25]. To test whether the results are robust for an extended warm period, we projected the three SOMs defined in July and August on May, June, and September (Fig. S4). Figure S5 shows the distribution of the three jet stream states among the different months of this extended period. May and June, primarily, and September to a lesser degree, are dominated by the mixed jet stream state. On the other hand, double jets are almost exclusively a July and August feature, as they hardly occur in the other months. Nevertheless, even when taking into account May-September (MJJAS), the composites of mean temperature and heatwave cumulative intensity (Fig. S4j-l and m-o) remain consistent with the ones seen in Fig. 2 for July-August only. Additionally, the significant upward trends in frequency and persistence of double jets (Fig. S4d-f) are also detected in the MJJAS analysis.In addition, although a lengthening of the mean European summer period has been observed[68,69] and often early summer heatwaves may have greater impacts on mortality[70], most of the severe European heatwaves are taking place in high-summer(July-August)[9]. For these reasons we decided to focus this analysis on July-August (JA) only, similarly to previous studies[71].

For the spatial domain, we tested both the whole hemisphere and the Eurasian sector only. The results were consistent (not shown), with double jets always showing up in one of the clusters with a similar frequency of occurrence, and therefore we chose to continue the analysis using the jet states over the Eurasian sector, as they are more relevant for heat extremes over Europe.

Therefore, SOMs were applied on the vertical zonal-mean zonal wind values of different pressure levels (from 800hPa up to 100hPa, at levels taken every 100hPa) of the daily ERA5 data of July and August for the period 1979–2020. Before applying the clustering algorithm, the zonal wind fields were weighted by the cosine of the latitude to account for the different size of the grid boxes. The three

jet stream states obtained by the SOMs are characterized by the composites of the days belonging to each of them, by a total frequency of occurrence for the whole period, and by their annual frequency and persistence. The annual frequency refers to the number of days that were clustered to each of those jet stream states (SOMs) per year and the annual persistence refers to the maximum number of consecutive days clustered to each of them per year.

Next, focusing on the most persistent double jet events, and in particular those with persistence exceeding the 90th percentile, we further clustered the meridional wind at 250hPa (v250) over Eurasia of those events in order to analyze dominant wave patterns. We clustered v250 in two SOMs and applied 10 random runs, choosing the one with the smallest quantization error. In order to check for the existence of trends in the occurrence of the two v250 clusters, we calculated a similarity index among the two v250 composites and the daily fields of v250. Then, the linear trend of those indices were estimated, and additionally, to account for non-linear trends a smoothed LOESS regression curve was applied (using a span of 0.75).All SOM implementations were done in R with the use of the latest version of the "kohonen" package[72].

**Links of jet stream states and climate.** Composites of different variables were made for the days belonging to each one of the three jet states described above. The composites of the zonal wind at the 250hPa level (u250) and of mean surface temperature were calculated for detrended and deseasonalized data and are presented in terms of anomalies from climatology, which was taken as the mean state of all days of the study period (July and August for 1979–2020). For the composites of heatwave cumulative intensity, we detrended the data by removing from each grid point the land mean trend of the Northern Hemisphere midlatitudes (25–70°N). The composites are then presented in terms of relative anomalies (%) compared to climatology.

In order to assess the links between jet stream states and heatwave metrics, we determined linear regression models to quantify the part of the observed heatwave variability explained by the jet stream in a linear model. To account for biases due to potential trends, we first detrended the regressors (jet state frequency and persistence) and the heatwave metrics (frequency and cumulative intensity).

Further, we calculated the estimated trends in the heatwave metrics at each grid point based on the linear regression model of the previous step. We used the regression parameters for the models calculated for the detrended data to estimate the heatwave trends using the non-detrended regressor (jet frequency and persistence). This way we can compare the estimated trend to the observed one. For the observed trend, we use the residual trend for each grid point after having subtracted the mean midlatitude-land trend. As a first-order approximation, we assume that the thermodynamical contribution to the increase in heatwaves is approximately similar for different midlatitude regions. The observed residual trend is the one that we can then attribute to other changes, such as dynamical changes in the jet stream and local feedbacks.

## Data availability

ERA5 datasets used in this study are publicly available by the European Center for Medium-Range Weather Forecasts (ECMWF).

## Code availability

All code used to produce the results and figures of this paper is available from the first author upon reasonable request.

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

## Acknowledgements

E.R. and G.B-.A. were supported by the German Federal Ministry of Education and Research within the ClimXtreme project (subprojects PERSEVERE, grant 01LP1901E, and NA2EE, grant 01LP1902F, respectively). This work used resources of the German Climate Computing Center (DKRZ) granted by its Scientific Steering Committee (WLA) under project ID bb1152 and of the high performance system at the Potsdam Institute for Climate Impact Research (PIK). K.K. was partially supported by the National Science Foundation (grant AGS-1934358). FL and DC acknowledge funding from the Netherlands Organization for Scientific Research (project PERSIST: Persistent Summer Extremes, grant 016.Vidi.171.011).

## Author contributions

E.R., D.C., K.K. conceived the idea and designed the study. E.R. performed all analysis, apart from the heatwave metrics that were calculated by G.B-.A., prepared the figures and wrote the first draft. E.R., D.C., K.K., G.B-.A. and F.L. contributed to the interpretation of the results and the final form of the manuscript.

## Funding

## Competing interests

The authors declare no competing interests.
