## [Peer Review File · Nature Communications]

Accelerated western European heatwave trends linked to more-persistent double jets over EurasiaReviewers' Comments:

Reviewer #1:

Remarks to the Author:

The authors investigated the upward trends of western European heatwaves that are three-to-four times faster compared to the rest of the northern midlatitudes over the past 42 years. A large part of this accelerated trend can be explained by atmospheric dynamical changes via an increase in the frequency and persistence of double jet stream states over Eurasia. I found the focus of the study very novel, which highlights the dynamic role of the double jet in the increased rate of European heatwaves. The overall approach of the study very interesting, and the topic is certainly a relevant one for the future extremes communities alike. The debate over changing midlatitude circulations, including slowing vs. accelerating of jets and strengthening vs. weakening of Rossby waves, is still going strong, and the target jet dynamics analysis conducted in this study is a very important part of resolving that debate. Although the double jet characteristics have already been mentioned by previous studies, the present manuscript highlights their dynamic roles, which could be a helpful reference to understand the high-impact extreme weather. However, I am concerned over the physical nature and implications of the double jet and why such a circumglobal zonal change in zonal wind alters such local extremes (why western Europe is special). Additional analysis is needed to clarify the midlatitude dynamics and physical mechanisms responsible for the double jet impacts. As such, I am recommending that the paper be resubmitted to the journal until the necessary evidence and analysis are provided.

Main Comments:

1. The motivation and the metric are good. However, the physical nature and characteristics of the double jet are not well described. For instance, there is a clear definition of jet streams in climatological fields, and the dynamical midlatitude jet over North Atlantic would be located in regions covering North Atlantic and Europe. Whether the southern branch of the jet stream in Figure 2b is a westerly anomaly or a subtropical jet stream? I recommend the authors add climatological jet stream contours over the u250 anomalies. The physical nature and possible causes of such double or mixed jet should also be further clarified or discussed.
2. Another concern is the interaction between zonal wind anomalies and meridional wind anomalies. It seems that the double jets are closely linked to the increased occurrence of western European heatwaves. Nothing also the maximum hot spot over eastern Europe, which is probably related to changes in blockings and associated meridional wind, as you mentioned in the introduction section. Therefore, I am curious about the links and interactions between zonal jet and meridional winds and large-scale circulation systems. The detailed wave-jet dynamics and physical mechanisms responsible for the double jet impacts should be clarified.
3. Fig. 3, the one case study is insufficient to demonstrate the double jet influence on heatwaves. Although the authors have conducted composite analysis of double jet days in Fig. 2, the one case study is not statistically convincing. There are belt-like negative anomalies of zonal wind over Europe and other midlatitudes. Why is western Europe special in the occurrence of heatwaves? What are the synoptic mechanisms for extreme weather?
4. Figs. 4-5. The double jet explains approximately 35% of the variance in western heatwaves. The estimated residual trend is comparative to the observed one when the mean midlatitude trend is subtracted. The authors claimed that the upward trend in the persistence of double jet events explains almost all of the accelerated heatwave trend in western Europe. Do you mean dynamical contributions? Why are the thermodynamic contributions given by the mean midlatitude trend? I suggest presenting evidence to support this viewpoint.
5. The last issue we are interested in is the future changes in jet streams and their contributions to the frequency and intensity of western European heatwaves. These analyses would be valuable for future predictions in the context of global warming. These results are also helpful to answer the question of whether the changes in jet streams are internal variability or response to global warming?
6. I recommend the authors to re-plot Fig. 4c. It would be easy to interpret if the X-Axis is changed to exact persistence days (rather than anomalies).

Reviewer #2:

Remarks to the Author:

Accelerated western European heatwave trends linked to more-persistent double jets over Europe by E. Rousi et al.

General comments

I recommend that this paper be published, if my main concerns are properly addressed.

The authors perform a study on how European temperature extremes are linked to large-scale atmospheric circulation and in particular with jet stream states. They analyze how potential changes there in might have contributed to upward heatwave trends. They also argued that the accelerated trend in western European heatwaves is linked to an increase in the frequency and persistence of double jets in the upper troposphere.

The introduction is very clearly written and makes you want to know in more detail what has been accomplished. I find the results interesting, however, I have some concerns in the methodology and about some unclear remarks:

Major comments:

My main concern is about the selection of months of the study (July-August). The authors have explained the exclusion criteria for June in section Methods, page 16, lines 390-398. However, It would be essential to see a similar figure to figure 2, at least in the Supplementary Material, for June or other selection of months (May-June or September) since May, June and/or September (Peña-Ortiz et al, 2015, <https://doi.org/10.1175/JCLI-D-14-00429.1>, , Sanchez-Benitez et al, 2018 <https://doi.org/10.1002/2018GL077253>), might be affected by important increases of temperatures specially in western Europe (dashed square in figure 4 and 5), the area where the authors have found the most relevant results .

Minor comments:

line 400: ...showing up....

lines 253-260: please include a reference.

Titles in figures S1 and S2 are the same. Please include some extra definitions (for instance 6 consecutive days in figure S1 and 3 consecutive days in figure S2) to make them different.

Detailed point-by-point response to reviewers

Reviewer #1:

The authors investigated the upward trends of western European heatwaves that are three-to-four times faster compared to the rest of the northern midlatitudes over the past 42 years. A large part of this accelerated trend can be explained by atmospheric dynamical changes via an increase in the frequency and persistence of double jet stream states over Eurasia. I found the focus of the study very novel, which highlights the dynamic role of the double jet in the increased rate of European heatwaves. The overall approach of the study very interesting, and the topic is certainly a relevant one for the future extremes communities alike. The debate over changing midlatitude circulations, including slowing vs. accelerating of jets and strengthening vs. weakening of Rossby waves, is still going strong, and the target jet dynamics analysis conducted in this study is a very important part of resolving that debate. Although the double jet characteristics have already been mentioned by previous studies, the present manuscript highlights their dynamic roles, which could be a helpful reference to understand the high-impact extreme weather. However, I am concerned over the physical nature and implications of the double jet and why such a circumglobal zonal change in zonal wind alters such local extremes (why western Europe is special). Additional analysis is needed to clarify the midlatitude dynamics and physical mechanisms responsible for the double jet impacts. As such, I am recommending that the paper be resubmitted to the journal until the necessary evidence and analysis are provided.

Response: We thank the reviewer for the positive and encouraging feedback on our study. We understand the concerns regarding the physical nature and implications of double jets for certain regions (and in particular western Europe). Therefore, we have done additional analysis which provides further insights to mentioned aspects, which are now discussed in more detail in the revised version of the Manuscript and in the following pages.

Main Comments:

1. The motivation and the metric are good. However, the physical nature and characteristics of the double jet are not well described. For instance, there is a clear definition of jet streams in climatological fields, and the dynamical midlatitude jet over North Atlantic would be located in regions covering North Atlantic and Europe. Whether the southern branch of the jet stream in Figure 2b is a westerly anomaly or a subtropical jet stream? I recommend the authors add climatological jet stream contours over the u250 anomalies. The physical nature and possible causes of such double or mixed jet should also be further clarified or discussed.

Response: We thank the reviewer for the thoughtful comment. We have added the climatological field of the zonal wind to Figure 2, panels a-c, as recommended, and also to panels g-l (see below). We are now describing the characteristics of the double jets (and the other jet states found) in more detail and put them into context with the climatological mean in the manuscript (lines 113-135). This shows that the southern branch of the double jet state seen in Figure 2b, which corresponds to a band of enhanced wind anomalies around 40°N in Figure 2h, is a strengthened and more confined subtropical jet stream and not a westerly anomaly, important for waveguidability. This is supported by the fact that the location of the positive anomalies in Figure 2h coincides with the climatological subtropical jet stream core (plotted with dashed contours in the same panel). This is in agreement with Figure 1 of White et al. (2021), which shows that the presence of the double jet structure over

Eurasia is seen in the extreme months but is absent in the climatology (Figure included below for reference). Moreover, while the driving mechanisms of subtropical and polar jet are different, they are often merged especially in summer (Molnos et al., 2017). Next to this, we have more firmly embedded our work into the existing literature discussing the potential underlying mechanisms, see Discussion (lines 313-371).

(revised) Figure 2. Jet stream states and surface temperature. (a-c) Composites of the vertical profile of the zonal (averaged over the Eurasian domain) mean zonal wind (u , shading) with frequency of occurrence provided in parenthesis. The climatological mean of the zonal mean zonal wind for the whole period is plotted with dashed contours (plotted from 5 to 20m/s every 5m/s). **(d-f)** Frequency (grey line) and maximum persistence (orange line) of each cluster per year. The decadal linear trend and respective p-values are given for both timeseries on the top left of the panels. **(g-i)** Anomaly composites of (linearly detrended) zonal wind at the 250hPa pressure level (u_{250}) for each cluster (shading). The climatological mean of the zonal wind at 250hPa is plotted with dashed contours (plotted from 5 to 25m/s every 5m/s). **(j-l)** Anomaly composites of (linearly detrended) mean surface temperature for each cluster. Anomalies in both cases are calculated with respect to daily climatology (to remove the seasonal cycle). **(m-o)** Composites of heatwave cumulative intensity (calculated after having removed the T_{max} mean midlatitude-land trend from each grid point) shown as relative anomaly (%) compared to the climatology. All figures refer to the months of July-August of the period 1979-2020.

Figure 1 from White et al. (2021).

2. Another concern is the interaction between zonal wind anomalies and meridional wind anomalies. It seems that the double jets are closely linked to the increased occurrence of western European heatwaves. Nothing also the maximum hot spot over eastern Europe, which is probably related to changes in blockings and associated meridional wind, as you mentioned in the introduction section. Therefore, I am curious about the links and interactions between zonal jet and meridional winds and large-scale circulation systems. The detailed wave-jet dynamics and physical mechanisms responsible for the double jet impacts should be clarified.

Response: Thank you for this comment. Indeed, double jets, despite their very zonal structure throughout the Eurasian sector, favor heatwaves in particular regions, including in western Europe. This is linked to pronounced anomalies in meridional winds. We did additional analyses showing dominant meridional wind states during double jets, see Figure 3. For this we composited the meridional wind speed at the 250hPa (v_{250} ; Figure 3a) for the most persistent double jet events (we choose to present the composite of those exceeding the 90th percentile of the double jet persistence, which is 11 days, but the results remain robust for different thresholds or even for all double jet days). As can be seen in Figure 3a, this composite pattern represents an amplified circumglobal wave (positive and negative anomalies coincide with the climatological-mean wave pattern). A similar amplified circumglobal wave pattern with the specific location has been found to be important for heat extremes in western Europe already in previous research (Kornhuber et al., 2019; Drouard et al., 2019; Kornhuber et al., 2020). Next we show that this composite originates from two preferred wave patterns. We clustered the meridional wind field of those persistent double jet events to see whether there are different preferred meridional wind patterns which might cancel each other out over some regions in a combined composite. Figure 3b and 3c shows the two clusters of v_{250} confirming this hypothesis. Thus, double jets are associated with amplified waves that come in two preferred positions over Eurasia that are phase-shifted by half a wavelength. SOM1 represents northerly winds over Scandinavia western Russia and southerlies over

the Ural Mountains (Figure 3b) creating heat extremes over Russia (Figure 3f). This pattern shows a statistically significant upward trend (Figure 3d), based on a similarity index of its composite with the daily v250 wind fields. In SOM2 this pattern is shifted to the west (Figure 3c) favoring heat extremes in western Europe (Figure 3g). The similarity index of SOM2 shows a small downward trend, which however turns to an upward trend when looking at the latest decade (2010-2020; Figure 3e). These European wave anomalies are part of a larger circumglobal wave pattern creating warm anomalies in western Siberia (SOM2) and eastern Siberia (SOM1). The anomaly composites of mean surface temperature for those two clusters (Figure 3d and 3e) thus show 4 hotspot regions (Russia and eastern Siberia for SOM1 and western Europe and western Siberia for SOM2), in agreement with the double jet composites of heatwaves in Figure 2n. We added relevant text about this to the Manuscript (lines 165-184, 331-371).

NEW Figure 3. Persistent double jets states. (a) Composite of meridional winds at 250hPa (v250) for double jet events exceeding the 90th percentile of double jet persistence (i.e., events lasting more than 11 consecutive days, see Table S1 for the 20 most persistent events and their duration; shading). Contour lines show the v250 July-August climatology for the whole period 1979-2020 (plotted from -5 to 8m/s every 2m/s). (b) (c) SOM cluster composites of v250 for persistent double jet events (frequency % of each SOM is shown in parenthesis). (d) (e) Time series of daily similarity index for the two SOMs of v250 winds. The red dashed line shows a linear regression fit, with its slope and p-value plotted on the top right. The continuous red line shows a smoothed LOESS curve fit (span of 0.75). (f) (g) Anomaly composites of (linearly detrended) mean surface temperature for each of the SOM clusters.

3. Fig. 3, the one case study is insufficient to demonstrate the double jet influence on heatwaves. Although the authors have conducted composite analysis of double jet days in Fig. 2, the one case study is not statistically convincing. There are belt-like negative anomalies of zonal wind over Europe and other midlatitudes. Why is western Europe special in the occurrence of heatwaves? What are the synoptic mechanisms for extreme weather?

Response: We thank the reviewer for this comment. We have now added 3 more case studies (1994, 2006 and 2018) representing particularly persistent double jet events to complement Figure 4 (see below) that give a more comprehensive picture of heatwave conditions during such persistent double jet events. All these summers are characterized by particularly persistent double jet events (see Table S1) and intense European heatwaves (Russo et al., 2015). As seen in the climatology of the wind at the 250hPa level (dashed contours in panels a/d/g/j) the western/central European region is normally coinciding with the exit region of the N. Atlantic jet stream. During double jets, this region is characterized by negative anomalies in zonal wind, while positive anomalies (corresponding to the 2 jets) can be seen to the north and to the south of the region. Additionally, as mentioned above, the double jets act as waveguides, enhancing certain wavenumbers in the meridional wind and favoring heatwaves in western Europe. This is also now evident from the new analysis we added on the meridional wind clusters of double jets, discussed above and in the manuscript. We updated the manuscript accordingly (see lines 197-221).

(Revised) Figure 4. Summers 1994, 2003, 2006, 2018: double jets and heatwave intensity. (a) Anomalies of the 250hPa zonal wind (u_{250} ; shading, anomalies from climatology; dashed contour lines show the total wind speed climatology plotted every 5m/s) for the longest double jet event in 1994: 23.07-19.08.1994. **(b)** Hovmöller diagram of the Eurasian (region seen in panel a) zonal mean

zonal wind anomalies for July-August 1994 (shading; 5day running means centered on each day from 01.07-31.08.1994). The vertical dashed line refers to the first day of August and the red horizontal lines on the time axis show the days identified by the SOMs as double jets (dates for 1994: 1-12.07, 23.07-19.08). **(c)** Spatial distribution of heatwave cumulative intensity for July-August 1994. **(d)** As in (a) but for 21.07-18.08.2003. **(e)** As in (b) but for July-August 2003 (dates of all double jets for 2003: 11.07, 16-19.07, 21.07-18.08, 24-31.08). **(f)** As in (c) but for July-August 2003. **(g)** As in (a) but for 11-30.07.2006. **(h)** As in (b) but for July-August 2006 (dates of all double jets for 2006: 11-30.07, 12-16.08, 31.08). **(i)** As in (c) but for July-August 2006. **(j)** As in (a) but for 04-25.07.2018. **(k)** As in (b) but for July-August 2018 (dates of all double jets for 2018: 04-25.07, 19-20.08, 25-31.08). **(l)** As in (c) but for July-August 2018.

4. Figs. 4-5. The double jet explains approximately 35% of the variance in western heatwaves. The estimated residual trend is comparative to the observed one when the mean midlatitude trend is subtracted. The authors claimed that the upward trend in the persistence of double jet events explains almost all of the accelerated heatwave trend in western Europe. Do you mean dynamical contributions? Why are the thermodynamic contributions given by the mean midlatitude trend? I suggest presenting evidence to support this viewpoint.

Response: Thank you for this thoughtful comment. As a first-order approximation, we assume that the thermodynamical contribution to the increase in heatwaves is approximately similar for different midlatitude regions. This builds on the first-order assumption that, the midlatitudes would respond uniformly to the external forcing imposed by anthropogenic global warming. In this we assume regional deviations in the response of temperature extremes to be driven by more complex physics including e.g. dynamical changes, see for example Wehrli et al. (2019), Suarez-Gutierrez et al. (2020), Stegehuis et al. (2021). Here we only quantify the role of double jets in shaping these differences in heatwave trends. We have made this more clear in the manuscript as well (see lines 285-302, 562-564).

5. The last issue we are interested in is the future changes in jet streams and their contributions to the frequency and intensity of western European heatwaves. These analyses would be valuable for future predictions in the context of global warming. These results are also helpful to answer the question of whether the changes in jet streams are internal variability or response to global warming?

Response: We agree with the reviewer that the next important topic to investigate are future changes in jet streams and their links to western European heatwaves. We also agree that this will help answer the question of whether the changes in double jets that we see in reanalysis data are due to internal variability or to external forcing, i.e. global warming (we added this in the Discussion, see lines 396-398). However, this is a substantial analysis that would exceed the scope of this particular paper and will be a better fit for a separate paper.

6. I recommend the authors to re-plot Fig. 4c. It would be easy to interpret if the X-Axis is changed to exact persistence days (rather than anomalies).

Response: Thank you for this recommendation. We have now changed the axis of Figure 4b and 4c (now Figure 5) in order to show double jet persistence in days and not anomalies (see below). For

consistency we did the same for double jet frequency and updated all respective plots in the SI figures as well (Figures S6, S7, S8).

(revised) Figure 5. Explained variance of heatwave cumulative intensity by double jet persistence.

(a) Explained variance (R^2) per grid point of heatwave cumulative intensity based on linear regression on double jet persistence. Statistically significant coefficients ($p < 0.05$) are marked with black dots. **(b)** Scatter plots of heatwave cumulative intensity anomalies aggregated over all land grid points of the extended European domain (as seen in panel a and in the red dashed box of Figure 1a and b) and double jet persistence. A linear fit (in red) and its confidence interval (dashed lines for the 5th and 95th percentiles obtained from 1000 bootstraps), R^2 (with 5th and 95th interval percentiles obtained from 1000 bootstraps in brackets), and p-values are shown on the top left of each plot. **(c)** As for (b) but with heatwave cumulative intensity aggregated only over land grid points with statistically significant coefficients in western Europe (dotted points in panel a within the region included in the dashed red box: 37-55°N and 9°W-14°E). The linear trend of the timeseries was removed before the regression was applied in all cases.

Reviewer #2:

General comments

I recommend that this paper be published, if my main concerns are properly addressed.

The authors perform a study on how European temperature extremes are linked to large-scale atmospheric circulation and in particular with jet stream states. They analyze how potential changes there in might have contributed to upward heatwave trends. They also argued that the accelerated trend in western European heatwaves is linked to an increase in the frequency and persistence of double jets in the upper troposphere. The introduction is very clearly written and makes you want to know in more detail what has been accomplished. I find the results interesting, however, I have some concerns in the methodology and about some unclear remarks:

Response: We thank the reviewer for their positive feedback, their interest and their recommendation to see this work published, eventually. In the following we provide a detailed response to the reviewer's comments.

Major comments:

My main concern is about the selection of months of the study (July-August). The authors have explained the exclusion criteria for June in section Methods, page 16, lines 390-398. However, It would be essential to see a similar figure to figure 2, at least in the Supplementary Material, for June

or other selection of months (May-June or September) since May, June and/or September (Peña-Ortiz et al, 2015, <https://doi.org/10.1175/JCLI-D-14-00429.1>, Sanchez-Benitez et al, 2018 <https://doi.org/10.1002/2018GL077253>), might be affected by important increases of temperatures specially in western Europe (dashed square in figure 4 and 5), the area where the authors have found the most relevant results .

Response: We thank the reviewer for this comment. We agree that heatwaves earlier or later in the season (such as in June and September) are also on the rise and have important impacts. We have now added an extra figure in the SI (Figure S4, also shown below) showing the respective results of Figure 2 but for an extended summer period, for the months of May-September (MJJAS). Additionally, Figure S5 shows the distribution of the jet stream states for each of those months. As seen in Figure S5, May and June, primarily, and September to a lesser degree, are dominated by the mixed jet stream state. On the other hand, double jets are almost exclusively a July and August feature, as they hardly occur in May, June or September. Nevertheless, even when taking into account MJJAS, the composites of mean temperature and heatwave cumulative intensity (Figure S4j-l and m-o) remain consistent with the ones seen in Figure 2 for July-August only. Of course, the climatological frequency of double jets now reduces to 17% of days as they almost exclusively occur during July-August days. Additionally, the significant upward trends in frequency and persistence of double jets (Figure S4d-f) are also detected in the MJJAS analysis. We added this sensitivity analysis to the Methods section along with more information about those differences between the rest of the months and July-August and we also included the suggested references (see lines 110-112, 505-517).

(NEW) Figure S4. Jet stream states and surface temperature for the extended summer period May-September (MJJAS). (a-c) Composites of the vertical profile of the zonal (averaged over the Eurasian domain) mean zonal wind (u , shading) with frequency of occurrence provided in parenthesis. The climatological mean of the zonal mean zonal wind for the whole period is plotted with dashed contours (plotted from 5 to 20m/s every 5m/s). (d-f) Frequency (grey line) and maximum persistence (orange line) of each cluster per year. The decadal linear trend and respective p-values are given for both timeseries on the top left of the panels. (g-i) Anomaly composites of (linearly detrended) zonal wind at the 250hPa pressure level (u_{250}) for each cluster (shading). The climatological mean of the zonal wind at 250hPa is plotted with dashed contours (plotted from 5 to 25m/s every 5m/s). (j-l) Anomaly composites of (linearly detrended) mean surface temperature for each cluster. Anomalies in both cases are calculated with respect to daily climatology (to remove the seasonal cycle). (m-o) Composites of heatwave cumulative intensity (calculated after having removed the T_{max} mean midlatitude-land trend from each grid point) shown as relative anomaly (%) compared to the climatology. All figures refer to the months of May-September of the period 1979-2020.

(NEW) Figure S5. Frequency of the three jet stream states for each of the months May-September.

Minor comments:

line 400: “showing up”

Response: Thank you for spotting this typo, it has been corrected.

lines 253-260: please include a reference.

Response: Thank you for this suggestion, we included two references describing the possible interactions of blocking with the jet stream (Steinfeld and Pfahl, 2019) and the role of advection of low PV air in maintaining blocking anticyclones (Pfahl et al., 2015) (lines 346-349).

Titles in figures S1 and S2 are the same. Please include some extra definitions (for instance 6 consecutive days in figure S1 and 3 consecutive days in figure S2) to make them different.

Response: Thank you for this comment, we added this information to the titles accordingly.

References

Drouard, M., Kornhuber, K. and Woollings, T.: Disentangling Dynamic Contributions to Summer 2018 Anomalous Weather Over Europe, *Geophys. Res. Lett.*, 46(21), 12537–12546, doi:10.1029/2019GL084601, 2019.

Kornhuber, K., Osprey, S., Coumou, D., Petri, S., Petoukhov, V., Rahmstorf, S. and Gray, L.: Extreme weather events in early Summer 2018 connected by a recurrent hemispheric wave pattern., *Environ. Res. Lett.*, 14(5), 054002, doi:10.31223/osf.io/tq23m, 2019.

Kornhuber, K., Coumou, D., Vogel, E., Lesk, C., Donges, J. F., Lehmann, J. and Horton, R. M.: Amplified Rossby waves enhance risk of concurrent heatwaves in major breadbasket regions, *Nat. Clim. Chang.*, 10(1), 48–53, doi:10.1038/s41558-019-0637-z, 2020.

Molnos, S., Mamdouh, T., Petri, S., Nocke, T., Weinkauff, T. and Coumou, D.: A network-based

detection scheme for the jet stream core, *Earth Syst. Dyn.*, 8(1), 75–89, doi:10.5194/esd-8-75-2017, 2017.

Pfahl, S., Schwierz, C., Croci-Maspoli, M., Grams, C. M. and Wernli, H.: Importance of latent heat release in ascending air streams for atmospheric blocking, *Nat. Geosci.* 2014 88, 8(8), 610–614, doi:10.1038/ngeo2487, 2015.

Russo, S., Sillmann, J. and Fischer, E. M.: Top ten European heatwaves since 1950 and their occurrence in the coming decades, *Environ. Res. Lett.*, 10(12), 124003, doi:10.1088/1748-9326/10/12/124003, 2015.

Stegehuis, A. I., Vogel, M. M., Vautard, R., Ciais, P., Teuling, A. J. and Seneviratne, S. I.: Early Summer Soil Moisture Contribution to Western European Summer Warming, *J. Geophys. Res. Atmos.*, 126(17), e2021JD034646, doi:10.1029/2021JD034646, 2021.

Steinfeld, D. and Pfahl, S.: The role of latent heating in atmospheric blocking dynamics: a global climatology, *Clim. Dyn.*, 53(9–10), 6159–6180, doi:10.1007/S00382-019-04919-6/FIGURES/12, 2019.

Suarez-Gutierrez, L., Müller, W. A., Li, C. and Marotzke, J.: Dynamical and thermodynamical drivers of variability in European summer heat extremes, *Clim. Dyn.*, doi:10.1007/s00382-020-05233-2, 2020.

Wehrli, K., Guillod, B. P., Hauser, M., Leclair, M. and Seneviratne, S. I.: Identifying Key Driving Processes of Major Recent Heat Waves, *J. Geophys. Res. Atmos.*, 124(22), 11746–11765, doi:10.1029/2019JD030635, 2019.

White, R. H., Kornhuber, K., Martius, O. and Wirth, V.: From Atmospheric Waves to Heatwaves: A Waveguide Perspective for Understanding and Predicting Concurrent, Persistent and Extreme Extratropical Weather, *Bull. Am. Meteorol. Soc.*, 1(aop), 1–35, doi:10.1175/BAMS-D-21-0170.1, 2021.

Reviewers' Comments:

Reviewer #2:

Remarks to the Author:

The manuscript has further improved after the revision. The authors have followed the suggestions of the reviewers, specifically, all my comments on the previous version of this manuscript have been addressed satisfactorily. Due to that, I recommend to accept the manuscript "Accelerated western European heatwave trends linked to more-persistent double jets over Eurasia" in the current form.